# Evaluating NHS Stop Smoking Service engagement in community pharmacies using simulated smokers: fidelity assessment of a theory-based intervention

Sandra Jumbe,[1,2] Wai Y James,[1] Vichithranie Madurasinghe,[3] Liz Steed,[1] Ratna Sohanpal,[4] Tammy K Yau,[5] Stephanie Taylor,[4] Sandra Eldridge,[4] Chris Griffiths,[3] Robert Walton[6]

This work has been previously presented as a thematic poster at the European Respiratory Society Milan 2017 International Congress.

For numbered affiliations see end of article.

**Correspondence to**
Dr Sandra Jumbe;
s.jumbe@qmul.ac.uk

## ABSTRACT

**Objectives** Smokers are more likely to quit if they use the National Health Service (NHS) Stop Smoking Service (SSS). However, community pharmacies experience low service uptake. The Smoking Treatment Optimisation in Pharmacies (STOP) programme aims to address this problem by enhancing staff training using a theory-based intervention. In this study, we evaluated intervention fidelity using simulated smokers (actors) to assess smoker engagement and enactment of key intervention components by STOP trained staff.

**Design** An observational pilot study.

**Settings** Five community pharmacies in North East London with an NHS SSS.

**Methods** Six actors, representative of East London's population, were recruited and trained to complete intervention fidelity assessments. Consenting pharmacy staff from five participating pharmacies received STOP Intervention training. Four weeks after the staff training, the actors visited the participating pharmacies posing as smokers eligible for smoking cessation support. Engagement behaviour by pharmacy staff and enactment of intervention components was assessed using a scoring tool derived from the STOP logic model (scoring range of 0–36), and contemporaneous field notes taken by actors.

**Results** 18 of 30 completed assessments were with STOP trained staff (10/18 were counter assistants). Mean score for smoker engagement was 24.4 (SD 9.0) points for trained and 16.9 (SD 7.8) for untrained staff, respectively. NHS SSS leaflets (27/30) were the most common smoking cessation materials seen on pharmacy visits. Most trained counter staff engaged with smokers using leaflets and a few proactively offered appointments with their cessation advisors. Appropriate use of body language was reported on 26/30 occasions alongside the use of key phrases from the STOP training session (n=8). Very few pharmacy staff wore STOP promotional badges (4/30).

**Conclusions** STOP training may change client engagement behaviour in pharmacy staff and could improve the uptake of the NHS SSS. A cluster randomised controlled trial is currently in progress to evaluate its effectiveness and cost-effectiveness.

### Strengths and limitations of this study

► We used simulated clients for a naturalistic fidelity assessment measuring enactment of a complex intervention to promote use of smoking cessation services in community pharmacies.

► The method enables quantitative and qualitative evaluation of pharmacy staff behaviour regarding client engagement by assessing whether key intervention materials are made available to service users, and rating what staff say and do.

► We found that this method worked well and gave an indication that important elements of the intervention were being enacted in the pharmacies; however, the lack of comparison data means we cannot necessarily attribute these findings to the Smoking Treatment Optimisation in Pharmacies intervention training.

► This study shows that the simulated client assessment method is feasible, and we will use this in the main trial to compare intervention and control pharmacies as part of the process evaluation.

**Trial registration number** ISRCTN16351033.

## BACKGROUND

Behavioural support for smoking cessation is highly cost-effective in reducing tobacco-related morbidity and mortality.[1–3] Behavioural interventions include advice, discussion and targeted activities aiming to: minimise motivation to smoke; increase resolve not to smoke; facilitate strategies to reduce exposure to smoking cues; improve management of smoking urges; and promote smoking cessation medication.[4–6] Rising prominence of evidence-based practice has resulted in increased implementation of behavioural support interventions as part of routine

healthcare.[7] One example of this is the National Health Service (NHS) Stop Smoking Service (SSS) in the United Kingdom (UK), which offers smoking cessation treatment including nicotine replacement therapy (NRT) to smokers trying to quit, alongside weekly consultations.[8] Results indicate that smokers engaged with this service are four times more likely to quit than those using NRT alone.[9]

Pharmacies with at least one staff member who has completed National Centre for Smoking Cessation and Training (NCSCT) training, often known as a stop smoking advisor (SSA), are able to deliver the NHS SSS.[10] There is strong evidence for the success of the pharmacy-led SSS in cost-effectiveness and good abstinence rates, endorsing behaviour change training of community pharmacy staff as an effective way of helping people to stop smoking.[11] However, service uptake in pharmacy settings is low.[12] While the recent decrease in smoking prevalence in the UK may be a factor, low uptake may also arise from the lack of awareness of pharmacies' public health role.[13] Studies also suggest low pharmacy staff confidence in their ability to deliver such services linked to the expectation of negative reactions from customers.[13–16]

Previous studies looking at the impact of smoking cessation training for pharmacists suggest a range of benefits including increased levels of counselling,[17 18] improved pharmacist consulting behaviour[19–21] and higher quit rates.[19 22] While previous interventions have shown benefits, their focus was primarily on the smoking cessation consultation itself rather than initial smoker engagement. The Smoking Treatment Optimisation in Pharmacies (STOP) programme was established to enhance delivery of the NHS SSS[23] by targeting self-efficacy, the motivation of pharmacy workers and skills to increase smoker engagement.[24]

This theory-based complex intervention was developed and tested for acceptability in 12 community pharmacies from three east London boroughs in an initial pilot study.[24] Specifically, 20 SSAs from these pharmacies attended two skills-based training sessions focused on communication and behaviour change skills. Study results confirmed the acceptability of the STOP intervention in terms of overall structure and face-to-face training content, with some participants reporting the use of newly learnt skills in practice. However, organisational barriers such as limited finances to cover pharmacist absence and poor acceptability of training venue and times limited pharmacy staff attendance. Another key finding was that very few staff members working at the pharmacy counter attended STOP training sessions. This limited effective service delivery because they were less able to engage in smoking-related conversation with clients.[24]

To address logistical barriers highlighted in the first pilot study for attending training, the study team conducted a focus group with four pharmacy-based SSAs and two counter staff. Based on feedback from this group, several changes were made to the intervention to enhance attendance and intervention uptake. First, training delivery was changed from two sessions to a half-day session on a Sunday morning with an option for on-site training on a weekday if more convenient to the pharmacy owner. Second, we took an organisational approach training all pharmacy staff (both NCSCT trained and untrained) within the same session to facilitate shared responsibility and focus on initial client engagement. The refined STOP Training Intervention content is summarised in table 1 which outlines the theories and behaviour change techniques[24 25] on which the intervention is based. The STOP logic model describing the programme theory for the intervention is shown in figure 1.

To address the receipt of the refined STOP intervention, we conducted a second pilot in November 2016. The focus of this second pilot study was to assess the fidelity of the refined STOP intervention, that is, to evaluate pharmacy staff performance of intervention skills in practice. Fidelity of intervention delivery involves assessing the extent to which core, prescribed intervention components are delivered as intended and are received by participants.[26] These two elements are critical for successful translation of evidence-based interventions into practice.[26 27] Results from fidelity assessments may highlight how the intervention is working and aspects that could be improved. However, fidelity assessment methods are not frequently reported.[26–28] In this study, we piloted the use of the 'simulated client' method as a way of assessing fidelity of the refined STOP intervention, with a focus on how well pharmacy staff engage with potential SSS clients.[29 30] Specifically, this paper reports on the potential of this method, to measure enactment of key elements of the intervention and evaluate impact of STOP training on client engagement into the SSS by pharmacy staff.

## METHODS
### Study design
This was an observational pilot study conducted in North East London Boroughs where simulated clients were used to examine how community pharmacies engage with pharmacy users asking for smoking cessation advice. We trained actors to play a particular scenario and tried to make them indiscernible from other service users.[24 29 30] This naturalistic method has been used in several previous studies to evaluate various aspects of services delivered by pharmacists.[29–33]

### Study procedure
#### Recruitment of pharmacies and STOP intervention delivery
Our primary aim was to pilot the use of the simulated client method as a way to assess the fidelity of the refined STOP intervention, with a focus on how well pharmacy staff engage with potential SSS clients. The training, therefore, focused on communication skills based on motivational interviewing and practising key phrases such as 'all quit attempts are a success', 'our service is free, delivered with an expert' and 'you can come back

**Table 1** Detailed description of the STOP intervention

| Pharmacy site initiation visit | Content | Theoretical basis * | Behaviour change techniques† |
|---|---|---|---|
| | Explain the study to the pharmacist in charge or manager<br>Mention potential revenue stream from smoking cessation<br>Emphasise to staff how this fits well with their wider role in health promotion<br>Raise awareness in all staff in preparation for the invitation to training.<br>Communicate the advantages of the STOP intervention over usual practice, that is, it is brief and show how it fits with overall 'pharmacy' identity<br>Address pre-implementation concerns<br>Provide financial incentive for attending training (only received on completion of training)<br>Emphasise backing from local and national opinion leaders and organisations (eg, Local Pharmaceutical Committee, Royal Pharmaceutical Society, local CCGs and public health commissioners) | Adoption by individuals: concerns in preadoption stage (DIT)<br>The innovation: compatibility; relative advantage; low complexity (DIT)<br>Outer context: incentives (DIT)<br>Diffusion and dissemination: opinion leaders (DIT) | 10.2 Material reward (behaviour)<br>9.1 Credible source<br>1.2 Problem solving<br>6.3 Information about others' approval |

| Training session | Content | Theoretical basis | Behaviour change techniques |
|---|---|---|---|
| Introduction | General orientation to STOP programme and aims of training.<br>Emphasise backing from local and national opinion leaders and organisations (eg, Local Pharmaceutical Committee, Royal Pharmaceutical Society, local CCGs and public health commissioners)<br>Discussion of the impact of advisor behaviour on client stop smoking outcomes so far and health benefits to patients from stopping smoking<br>Delivered in mixed groups of pharmacists and other pharmacy workers to promote cohesive-working practices within the individual pharmacies | Outcome expectancies (SCT)<br>Diffusion and dissemination: opinion leaders (DIT)<br>Implementation and routinisation: organisational structure (DIT) | 5.1 Information on health consequences of behaviour<br>9.1 Credible source<br>10.6 Non-specific incentive<br>15.1 Verbal persuasion about the capability |
| Why are we here? | Smoking facts and exploration of motivation for helping smokers to quit with feedback<br>Discuss focus on pharmacy setting, emphasising the non-medication related, professional and public health aspects of the pharmacy role<br>Does engaging and supporting smokers' quit fit with role identity, any barriers? Encourage self-perception as supporters and providers of health, how one will feel if help smokers quit<br>Emphasise the non-medication related, professional and public health aspects of the pharmacy role, promote a person-centred rather than product-centred ethos and foster a strong sense of professionalism | Intrinsic and extrinsic motivators (SDT)<br>The innovation: compatibility (DIT) | 5.6 Information about emotional consequences<br>9.2 Pros and cons<br>6.3 Information about other approval<br>13.1 Identification of self as a role model<br>15.3 Focus on past success |
| Engaging clients | Celebrate successful cases<br>Group exercise and discussion on difficult and easy clients to engage—potential problems and solutions<br>Addressing pharmacy workers beliefs and attitudes, for example, prejudgement of success or failure | Self-efficacy (SCT)<br>Modelling (SCT)<br>Vicarious learning (SCT) | 1.2 Problem solving<br>8.1 Behavioural practice and rehearsal<br>9.2 Pros and cons<br>13.3 Incompatible beliefs<br>15.3 Focus on past success |
| Patient-centred approach: building rapport and shift of focus | Introduction of patient-centred approach using group exercise. Group identification and discussion of the importance of utilising basic communication skills (rapport, active listening, questioning)<br>Review how patient-centred approach can be incorporated into smoking cessation interactions for better patient outcomes<br>Role play demonstration with senior pharmacist, participant practice. How to maximise opportunity with environmental resources, for example, staff wearing STOP badges to prompt client interaction, STOP posters<br>Emphasise predictable improved results, simplicity of use and benefits over the usual practice | Self-efficacy (SCT)<br>Modelling (SCT)<br>Vicarious learning (SCT)<br>The innovation: relative advantage; compatibility; low complexity (DIT) | 4.1 Instruction on the performance of behaviour,<br>6.1 Demonstration of behaviour<br>7.1 Prompts and cues<br>8.1 Behavioural practice and rehearsal<br>8.2 Behaviour substitution<br>8.6 Generalisation of target behaviour<br>9.2 Pros and cons<br>9.3 Comparative imagining of future outcomes |
| NCSCT knowledge review | Review the group's NCSCT knowledge with a quiz and general feedback | Self-efficacy (SCT) | 1.6 Discrepancy between current behaviour and goal<br>9.1 Credible source |
| Pharmacy role in smoking cessation | Discuss individual pharmacies' NHS Stop Smoking Service structure and purpose of smoking treatment, with experienced advisers sharing current and best practice. Group reflection on challenges | Homophily (DIT) | 1.7 Review outcome goal(s) |
| Behaviour change as smoking cessation treatment | Emphasise behaviour change support as part of smoking cessation treatment within NHS SSS<br>Information on how to assess someone's readiness to quit smoking using 1–10 scales | Self-efficacy (SCT)<br>Self-regulation (SCT)<br>The innovation: compatibility (DIT) | 4.1 Instruction on how to perform behaviour<br>8.1 Behavioural practice and rehearsal |

Continued

**Table 1** Continued

| Training session | Content | Theoretical basis | Behaviour change techniques |
|---|---|---|---|
| Behaviour change using patient-centred approach in smoking cessation treatment: double whammy | Brainstorm factors that influence behaviour change—(role of beliefs, capability, opportunity alongside knowledge). How to elicit individuals' motivations, barriers and potential strategies to change behaviour versus offering solutions. Using 'What else questions'. Understanding the 'non-smoker identity' and how to communicate with a client. Demonstration and roleplay. What makes this client-centred approach difficult—advantages, disadvantages, barriers and strategies to aid implementation | Outcome expectancies (SCT); Modelling (SCT); Self-efficacy (SCT); The innovation: fuzzy boundaries (DIT) | 1.2 Problem solving; 4.1 Instruction on how to perform behaviour; 6.1 Demonstration of behaviour; 8.1 Behavioural practice and rehearsal; 9.2 Pros and cons |
| Client engagement in pharmacy settings: planning a quit and dealing with lapses | Discuss planning a quit and how to help people make a specific plan using a SMART approach. Go over ways to discuss with lapses and provide supportive praise. Discussion of how to talk about willpower and the role of the open door. Watch and reflect on video of strong and weak consultations of quit planning. Demonstration and roleplay | Modelling (SCT); Self-efficacy (SCT) | 1.1 Goal setting (behaviour); 1.2 Problem solving; 3.1 Social support (unspecified); 4.1 Instruction on how to perform behaviour; 6.1 Demonstration of behaviour; 8.1 Behavioural practice and rehearsal |
| Client engagement in pharmacy settings: goal setting and making a commitment | Facilitate goal setting and elicit verbal commitment from participants. Demonstration (via video). Practice cohesive working among trainees through role play using multiple scenarios and observer feedback | Modelling (SCT) | 1.1 Goal setting (behaviour); 1.9 Commitment; 3.1 Social support (unspecified); 8.1 Behavioural practice and rehearsal; 15.1 Verbal persuasion about the capability |
| Implementing STOP | Review how to implement STOP in practice (ie, prompts and WhatsApp support) by facilitating discussion of implementation plans alongside facilitators or barriers with the pharmacy team. Highlight use of local champions and prompts/cues including the Double Whammy (a desk calendar with visual cues and example questions to ask) to prompt client interaction. Highlight ongoing social support via WhatsApp. Promote adaptation of non-core elements of the intervention through a prompted pharmacy team meeting to discuss the implementation of the intervention according to the needs of each individual pharmacy, for example, appointment of individual champions, monthly 'STOP' smoking days | Self-regulation (SCT); Intrinsic/Extrinsic motivators (SDT); The innovation: augmentation/support (DIT); The innovation: trialability; reinvention; fuzzy boundaries; champions (DIT); Implementation and routinisation: organisational structure (DIT) | 4.1 Instruction on how to perform behaviour; 1.2 Problem solving; 7.1 Prompts and cues; 3.2 Social support (practical); 1.4 Action planning; 10.1 Material incentive (behaviour) |
| End of session | Participants provided with a certificate for attending the training linked to CPD (endorsed by RPS). Provide a financial reward for those who have completed intervention training | Outer context: incentives (DIT) | 10.2 Material reward |

| 6 Week booster visit | Content | Theoretical basis | Behaviour change techniques |
|---|---|---|---|
| Feedback on smoker engagement and update on STOP implementation | Identification of organisational barriers, facilitators to implementing STOP in individual pharmacies. Facilitating action plans to implement STOP in their pharmacy. Any further thoughts on how the intervention can be adapted to local circumstances? Review pharmacy staffs' self-efficacy of skills. Troubleshoot based on performance feedback, assessment by simulated clients* and staff's self-reported self-efficacy. *Simulated clients are trained actors who approach staff in intervention pharmacies using smoking-related scenarios to assess client engagement and evaluate the presence of STOP smoking environmental cues (posters, badges) | The innovation: fuzzy boundaries (DIT); Adoption by individuals: concerns in preadoption stage (DIT); The innovation: reinvention; The innovation: augmentation/support (DIT); Self-efficacy (SCT); The innovation: trialability (DIT) | 1.1 Goal Setting; 1.2 Problem solving; 1.4 Action Planning; 1.5 Review behaviour goal(s); 1.6 Discrepancy between current behaviour and goal; 1.7 Review of outcome goal; 1.9 Commitment; 2.2 Feedback on behaviour; 2.7 Feedback on the outcome of behaviour; 7.1 Prompts and cues; 8.3 Habit formation |

See Bandura A. Health promotion from the perspective of social cognitive theory. Psychology and health. 1998 Jul 1;13(4):623–49.
*See Greenhalgh T, et al. Diffusion of innovations in service organisations: systematic review and recommendations. The Milbank Quarterly. 2004 Dec;82(4):581–629.
†Derived from the Behaviour Change taxonomy (Michie S. et al The behaviour change technique taxonomy (v1) of 93 hierarchically clustered techniques: building an international consensus for the reporting of behaviour change interventions. Annals of behavioural medicine. 2013 Mar 20;46(1):81–95.
CPD, Continuing Professional Development; DIT, Diffusion of Innovation Theory; NCSCT, National Centre for Smoking Cessation Training; RPS, Royal Pharmaceutical Society; SCT, Social Cognitive Theory; SDT, Self-determination theory; STOP, Smoking Treatment Optimisation in Pharmacies.

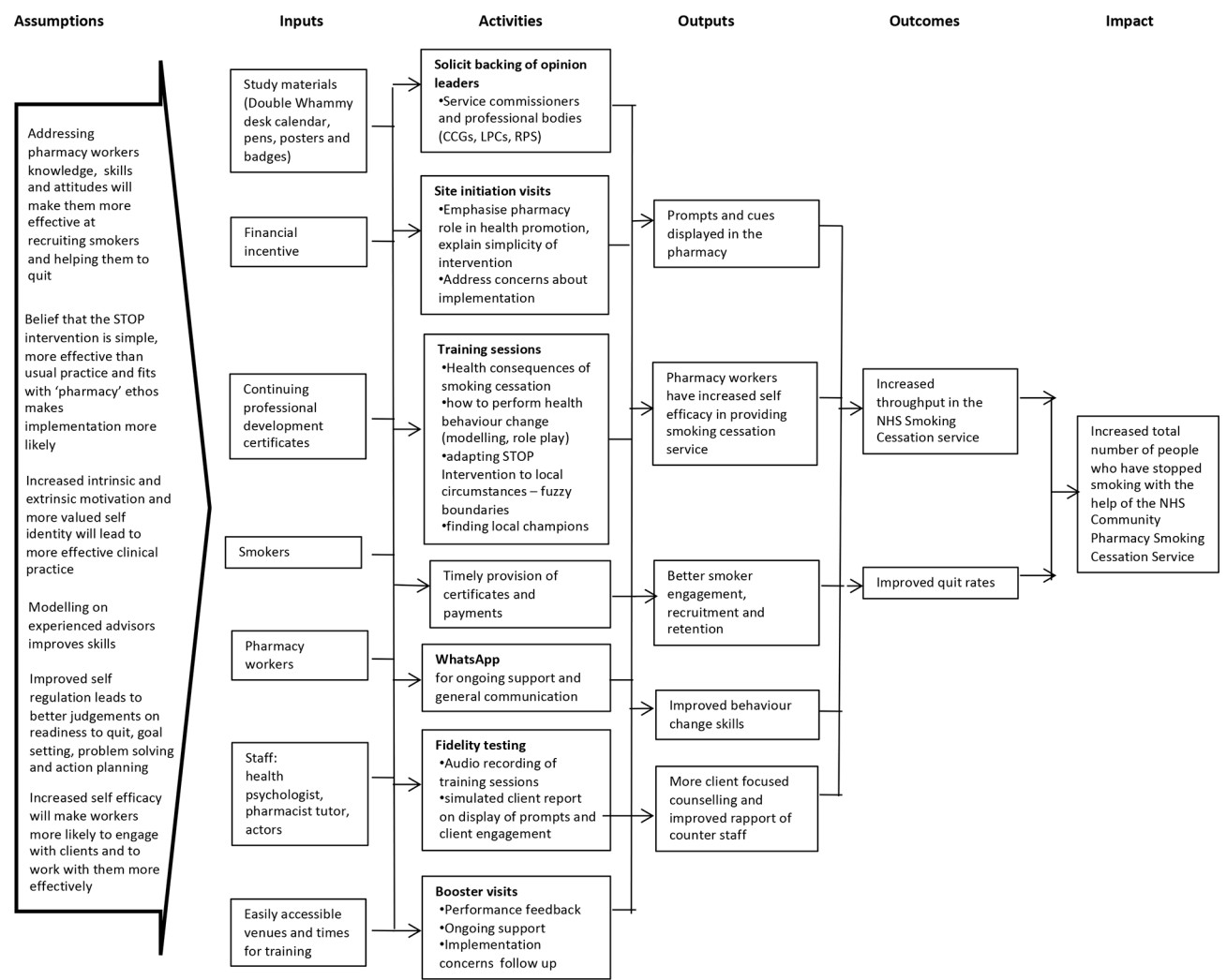

**Figure 1** STOP programme final logic model. CCG, Clinical Commissioning Group; LPC, Local Pharmaceutical Committee; NHS, National Health Service; RPS, Royal Pharmaceutical Society; STOP, Smoking Treatment Optimisation in Pharmacies.

anytime for support', to facilitate better engagement with potential SSS clients.[24]

The STOP team contacted 15 community pharmacies in North East London Boroughs commissioned to deliver the SSS and recruited 6 pharmacies. Based on our previous pilot experience and qualitative work, this sample size was deemed appropriate to assess intervention fidelity.[16 24] One pharmacy subsequently dropped out due to a family emergency. Table 2 outlines the characteristics of the 20 staff members from the five participating

| Table 2 | Pharmacy staff demographics | | |
|---|---|---|---|
| **Characteristics** | **Support staff (n=16)** | **Stop smoking advisors (n=4)** | **Total** |
| Mean age, years (range) | 29 (16–49) | 37 (30–54) | 30 (16–54) |
| Male (%) | 38 | 100 | 50 |
| Graduate or higher (%) | 25 | 100 | 40 |
| Never smoked (%) | 75 | 100 | 80 |
| *Job titles* | | | |
| | Counter assistant | 10 | |
| | Dispensing chemist | 4 | |
| | Trainee pharmacist | 2 | |
| | Pharmacist | | 2 |
| | Pharmacist technician | | 1 |
| | Business manager | | 1 |

**Table 3** Simulated clients demographics

| ID | Age | Gender | Ethnicity | Education | Smoking status |
|----|-----|--------|-----------|-----------|----------------|
| 01 | 49 | Male | White | Postgraduate | Never smoked |
| 02 | 56 | Female | White British | Other | Ex-smoker |
| 03 | 54 | Male | Mixed | Graduate | Ex-smoker |
| 04 | 32 | Male | Black | Graduate | Never smoked |
| 05 | 22 | Female | Mixed | NVQ L3 | Ex-smoker |
| 06 | 58 | Female | White British | Professional (CPCAB) | Ex-smoker |

pharmacies who individually gave their consent and attended STOP training. All staff were reimbursed for travel expenses and their time with amounts based on participants' usual day rates. Finally, a WhatsApp group was set up for participating staff as a communication tool and an information-sharing platform.

### Recruitment of simulated clients

Soon after the STOP training was delivered, the STOP team identified and recruited six actors for training to assess pharmacy worker engagement with clients and to record display of smoking cessation materials while posing as simulated clients. These simulated clients were purposefully sampled to represent diverse backgrounds reflecting the east London population[34] thus minimising the risk of detection by pharmacy staff (table 3).

### Simulated client training

All simulated clients attended a 1.5-hour training session led by the STOP team. During the training, each simulated client was assigned one of six smoking-related scenarios (figure 2), which they practised during the training session and received group feedback. The session also involved demonstrating the use of the fidelity assessment questionnaire, developed by the STOP team to assess pharmacy staff smoker engagement behaviour (figure 3). The fidelity assessment involved noting the presence of stop smoking-related material in the pharmacy environment and rating the

extent to which pharmacy staff build general rapport and the conversation related to the NHS SSS with the simulated client, using a Likert scale (figure 3). The questionnaire has a scoring range of 0–36, where higher ratings indicate better client engagement. Items on the questionnaire were chosen to evaluate outputs of the activities represented in the STOP logic model (figure 1). Simulated clients also provide written feedback on the questionnaire, referred to as field notes, to detail relevant aspects of their interaction where necessary. We used the Qualtrics online survey software[35] enabling simulated clients to complete their fidelity assessments electronically after each visit and aiding immediate receipt of source data for the study team.

The simulated clients practised completing the fidelity assessment questionnaire through Qualtrics; they observed the trainers role-playing interactions between smokers and pharmacy staff and rated each interaction using Qualtrics. In addition, simulated clients were asked to note the name of each pharmacy worker they interacted with by asking directly or looking at their name badge. If this was not possible, they provided a detailed description of the pharmacy worker instead. This enabled the study team to identify simulated interactions that were with pharmacy staff who had attended the STOP intervention training as not all staff in participating pharmacies attended the training. Simulated clients were not aware of the training status of the pharmacy staff they assessed. Therefore, these were blind outcome assessments.

### Fidelity assessment pharmacy visits

Between 4 and 6 weeks after STOP intervention training, each simulated client visited every pharmacy once using their simulated smoker–client scenario, resulting in 30 completed fidelity assessments (6 actors × 5 pharmacies). Simulated clients were given 2 weeks to complete their visits; to facilitate a naturalistic approach, no fixed schedule was imposed. All participating pharmacies provided written consent for the visits. However, staff were blind to visit timelines and frequency to minimise risk of detection and resulting change in their usual consulting behaviour. Simulated clients received £30 for every completed fidelity assessment.

| Scenario 1 | Scenario 2 | Scenario 3 |
|------------|------------|------------|
| Individual picks up SSS visual resource and reads through, attends counter and asks if the pharmacy offers advice on how to stop smoking. Has discovered teenage child smokes and wants them to stop.<br><br>Actor required: middle aged woman, white | Individual attends counter coughing, with cigarettes obviously on display e.g. holding in hand and putting on counter. Individual tells pharmacy staff they have asthma and need advice on how to stop smoking.<br><br>Actor required: middle aged male, white | Individual attends counter with non-smoking related request e.g. purchase pregnancy vitamins. Has cigarettes visible and obviously on display e.g. holding in hand and putting on counter.<br><br>Actor required: middle aged male, ethnic minority |
| Scenario 4 | Scenario 5 | Scenario 6 |
| Individual attends counter wheezing, with asthma pump e.g. holding in hand and putting on counter. Individual tells pharmacy staff they have asthma and COPD but still smoke. Asthma is getting worse so they need advice on how to stop smoking.<br><br>Actor required: middle aged female, white | Individual attends the pharmacy counter, complaining of a sore throat and need of lozenges. Explains that they went partying last night with friends and was smoking so maybe sore throat is linked to that. Friend and mother nagging that sore throat is due to smoking. Ask pharmacy worker what they think it could be?<br><br>Actor required: young female, ethnic minority | Individual attends counter, complaining of stress from girlfriend/mum to stop smoking because he has been getting sore throats a lot lately. Not sure if he wants to quit but thought he should check out what the pharmacy have to offer. Saw a sign that says pharmacy has a Stop Smoking Service.<br><br>Actor required: young male, ethnic minority |

**Figure 2** Smoking-related scenarios. COPD, chronic obstructive pulmonary disease; SSS, Stop Smoking Service.

Actor ID:   ___ ___

**Fidelity Assessment Questionnaire**

| Date:<br>DD / MM / YEAR | Time:<br>__:__ hours | Scenario number: | Pharmacy ID |
|---|---|---|---|
| | | | |

*Environment*

| Were the following visible within the pharmacy:- | *Tick if seen* |
|---|---|
| NHS Stop Smoking Service poster or audiovisual information | |
| NHS Stop Smoking Service leaflets | |
| STOP Study Poster | |
| Staff wearing STOP Study Badges | |

*Rapport*

To what extent did the pharmacy worker demonstrate:-        (circle a number)

a) good body language e.g. eye contact, attentiveness        0  1  2  3
b) good listening skills e.g. focused on conversation        0  1  2  3
c) use of open questions        0  1  2  3
d) picking up on client's verbal or visual cues        0  1  2  3

*Conversation*

To what extent did the pharmacy worker:-        (circle a number)

a) Initiate conversation of smoking in response to a verbal or visual prompt        0  1  2  3
b) Raise smoking directly e.g. do you smoke        0  1  2  3
c) Raise smoking indirectly e.g. by talking about second hand smoke        0  1  2  3
d) Tell you there is a smoking cessation service available in the pharmacy        0  1  2  3
e) Highlight that the service is free aside from NRT if pay for prescriptions        0  1  2  3
f) Highlight that people who use the service are 4 times more likely to quit        0  1  2  3
h) Ask you whether you want a referral to the service        0  1  2  3
i) Close conversation by saying you can come back anytime for help/support        0  1  2  3

| Additional comments: | Total score: |
|---|---|
| | |

**Figure 3** Fidelity assessment questionnaire. NHS, National Health Service; NRT, nicotine replacement therapy; STOP, Smoking Treatment Optimisation in Pharmacies.

## Data analysis

Data collected on Qualtrics were analysed using SPSS V.24. Descriptive statistics were used to analyse pharmacy staff recruitment and fidelity assessment outcomes. The simulated clients' field notes were also examined quantitatively, particularly focusing on the number of times smoking cessation materials, and specific actions or phraseology from the STOP training was reported on. Quotes from the field notes were used to exemplify the quantitative data generated by the fidelity assessment tool.

## Patient and public involvement

No patients and/or public were involved in the development or conduct of this second pilot study. However, we consulted an expert group of pharmacy workers before this study who advised on potential barriers to engaging with STOP intervention.

## RESULTS
### Smoker engagement ratings

All simulated clients completed their allocated five visits. In total, 18 assessments were with pharmacy staff (8 pharmacists or stop smoking advisors, and 10 counter assistants) who had attended the STOP training and 9 assessments were with pharmacy staff who had not attended STOP training. On three occasions, the training status of the assessed pharmacy worker was unclear.

**Table 4** Client engagement ratings from simulated smokers

| | | Client engagement ratings by simulated clients | | | | | | Average score |
|---|---|---|---|---|---|---|---|---|
| | | 1 | 2 | 3 | 4 | 5 | 6 | |
| S06 | 100 | 36 | 34 | 36 | 34 | 34 | 29 | 35 |
| S02 | 60 | 29 | 14 | 18 | 29 | 6 | 14 | 18 |
| S01 | 57 | 18 | 14 | 16 | 25 | 27 | 6 | 18 |
| S05 | 40 | 10 | 4 | 20 | 15 | 28 | 14 | 15 |
| S03 | 33 | 14 | 24 | 25 | 27 | 22 | 20 | 22 |
| Pharmacy site | % of pharmacy staff trained | | | | | | | |

STOP, Smoking Treatment Optimisation in Pharmacies.

◻ Completed STOP training

◼ Did not attend training

◼ Unable to identify

Table 4 shows total smoker client engagement scores allocated to each pharmacy interaction. Greyscale colours are used to differentiate between interactions with pharmacy staff who attended the STOP intervention training and those who did not. Simulated clients encountered more STOP trained staff in pharmacies with a high staff STOP training attendance. Pharmacies with a higher proportion of trained staff tended to have higher client engagement scores. For example, S06 was the only pharmacy where all staff attend STOP training was given the highest score by all actors. In contrast, S05 where 40% of its staff attended STOP training had the lowest average client engagement score. Simulated clients rated interactions with STOP trained staff higher than interactions with staff who did not attend STOP training (table 5).

Table 6 shows the average ratings each simulated client gave to the five participating pharmacies as part of the fidelity assessment, including display of smoking cessation materials and communication skills. Scores for client rapport indicate that pharmacy staff demonstrated very good or good use of body language and good listening skills. However, the use of open questions was limited. In terms of specific smoking-related conversation, pharmacies were given low ratings for indirectly raising the topic of smoking and for highlighting strong evidence for high SSS quit rates. Scores for directly raising the topic of smoking were moderate to low. However, pharmacy staff were rated highly for telling clients about the SSS, the fact that it was heavily subsidised or free for those who do not pay for their prescriptions and informing clients that they could come back for smoking cessation support at any time.

**Table 5** Mean client engagement scores for trained vs untrained staff

| Attended STOP training | N | Mean | SD |
|---|---|---|---|
| Overall client engagement | | | |
| No | 12 | 16.9 | 7.7 |
| Yes | 18 | 24.4 | 9.0 |
| Building rapport | | | |
| No | 12 | 6.7 | 3.7 |
| Yes | 18 | 9.1 | 2.3 |
| Conversation | | | |
| No | 12 | 10.3 | 5.1 |
| Yes | 18 | 15.3 | 7.1 |

STOP, Smoking Treatment Optimisation in Pharmacies.

## DISPLAY OF STOP SMOKING MATERIALS

Table 6 shows how many times the simulated clients (n=6) reported seeing specific smoking cessation materials during their five pharmacy visits. The materials were signposted in the STOP training as useful resources for engaging potential smoker-clients to NHS SSS. At least four or more of six simulated clients reported seeing NHS SSS leaflets during their pharmacy visit. Most of the pharmacy staff that the simulated clients encountered did not wear a STOP badge.

### Field notes taken by simulated clients

Quantitative analysis of the simulated clients' field notes provided an indication of partial enactment of the STOP Intervention. For example, NHS SSS leaflets were a popular tool for disseminating smoking cessation information and all simulated clients reported being given a leaflet from at least one pharmacy (table 6). This was also evident in the simulated client comments, with some pharmacies providing customised leaflets, indicating efforts to deliver a more tailored/personalised service.

**Table 6** Average simulated client fidelity assessment ratings including display of smoking cessation materials and communication skills

|  | SC 1 | SC 2 | SC 3 | SC 4 | SC 5 | SC 6 |
|---|---|---|---|---|---|---|
| **Display of smoking cessation materials** | | | | | | |
| NHS SSS poster/audio information | Y=3 | Y=1 | Y=0 | Y=5 | Y=0 | Y=2 |
| NHS SSS leaflets | Y=4 | Y=5 | Y=3 | Y=5 | Y=5 | Y=5 |
| STOP study poster | Y=1 | Y=1 | Y=2 | Y=3 | Y=0 | Y=3 |
| STOP study badge | Y=0 | Y=1 | Y=1 | Y=1 | Y=0 | Y=1 |
| ***Rapport with clients** | | | | | | |
| Good use of body language | 2 | 2 | 2 | 3 | 3 | 2 |
| Good listening skills | 2 | 2 | 2 | 3 | 3 | 2 |
| Use of open questions | 1 | 1 | 2 | 1 | 2 | 2 |
| Picking up client's verbal or visual cues | 2 | 2 | 2 | 2 | 3 | 2 |
| ***Conversation** | | | | | | |
| Initiate conversation on smoking in response to cues | 2 | 1 | 2 | 3 | 2 | 2 |
| Raise smoking directly | 2 | 1 | 1 | 2 | 2 | 0 |
| Raise smoking indirectly | 1 | 0 | 1 | 1 | 0 | 0 |
| Tell client about available SSS in pharmacy | 2 | 3 | 3 | 3 | 2 | 2 |
| Highlight free or subsidised service | 2 | 3 | 3 | 2 | 2 | 1 |
| Highlight facts on SSS high quit rates | 1 | 1 | 1 | 1 | 1 | 1 |
| Ask client about service referral | 2 | 1 | 3 | 2 | 2 | 1 |
| Close conversation with 'come back anytime' for help | 2 | 2 | 2 | 3 | 2 | 2 |

'Y' refers to the total number of times a simulated client ticked to confirm the display of a smoking cessation material from their five pharmacy visits.
*Numbers here refer to average client engagement scores assigned by simulated smokers across their five pharmacy visits. Range of 0–3, where 0 indicates no rapport or conversation and 3 indicates very good rapport or conversation.
NHS, National Health Service; SC, simulated client; SSS, Stop Smoking Service.

I was surprised to be given a leaflet with his name printed on it and the times the pharmacy is open and the leaflet did have smoking cessation clinic on it (actor 6).

Gave out loads of standard (NHS) leaflets plus one he made up himself (actor 1).

Aside from leaflets, actors reported being given key details of the NHS SSS proactively in interactions with STOP trained staff. This was at times despite there being other customers in the pharmacy waiting to be served.

Chemist (said) we can help you if you decide to go on the 12-week course with emotional help and medication (actor 2).

Stressed that I can come anytime for a consultation with himself, it is free, you can just walk in, takes about 15–20 min gave me a leaflet and took the time to go through the 12-week smoking cessation programme despite the pharmacy being busy (actor 5).

There was also evidence of clients being offered a smoking cessation appointment by staff at the counter. Simulated clients found that most pharmacy staff were proactive in giving the option to book an appointment to see the smoking cessation specialist. A few actors were

directed to a private room where they spoke with an advisor about joining the NHS SSS.

He did mention that it was a 12-week programme and offered to sign me up a few times (actor 1).

Very helpful. Offered me the programme immediately and nicotine replacement therapy (actor 6)

Staff at counter were very eager. It was hard to stop them from referring me to their 'trained' person for a private chat (actor 4).

Even when actors declined to join the service, pharmacy staff were keen on giving them smoking cessation leaflets to take home with them or telling clients to come back whenever they felt ready.

Told me about the 12-week programme and when I said I had to leave, said I could come back anytime I felt more ready. Got leaflets too (actor 4).

These two aspects of giving SSS leaflets and the 'keep the door open' approach were focused on during the training sessions, as several attendees raised the issue of missing potential clients due to the busy pharmacy environment. Throughout group discussions and role-play exercises, the group felt this approach would particularly

help counter staff minimise loss of potential clients needing smoking cessation support.

Several simulated clients described evidence of trained pharmacy staff trying to build rapport by using body language and active listening.

> Good eye contact, very pleasant (actor 2).
>
> Very attentive and listened to content in my scenario (actor 4).

On one occasion, one staff member even shared their personal experience of using the service, suggesting a sense of personal commitment to helping others quit.

> Self-disclosed that she is using the service and has found it amazing (actor 6).

Simulated clients also reported the use of key phrases or facts by pharmacy staff that were covered in the STOP training sessions or circulated on the WhatsApp group during their smoking cessation interactions, demonstrating retention of knowledge from the STOP training intervention.

> Tom was eager to tell me that my son would need to want this himself … he said there would be little or no point if my son did not want to stop himself (actor 6).
>
> 70% of smokers want to quit but just need help (actor 5).

## DISCUSSION

This pilot study suggests that the methods we designed using simulated clients to assess the fidelity of a complex intervention worked in practice and gives preliminary evidence of enactment of key intervention components. Findings also indicate that client engagement was better for pharmacy staff who attended STOP training, with improved consulting styles and increased use of intervention materials; quantitative analysis of contemporaneous field notes taken by simulated clients confirmed the availability and use of some smoking cessation materials. The analysis also suggested that pharmacy staff (including those without NCSCT training) were using consultation skills and appropriate words and phrases which were taught in their STOP intervention training. From a social cognitive theoretical perspective,[36] this potentially demonstrates improved knowledge through vicarious learning and increased self-efficacy to provide basic SSS information to clients.

### Strengths

Simulated clients, commonly known as 'mystery shoppers', are widely used in marketing to measure aspects of customer care. As staff are unaware of the simulated client's identity, this provides an opportunity for a naturalistic fidelity assessment of how well knowledge and skills from the intervention were received. Similar methods

have been found to be rigorous and robust for measuring practice in this setting.[29 31 32]

The fidelity assessment questionnaire used by the actors allowed for both quantitative and qualitative measurement of pharmacy staff behaviour regarding engagement with smoking cessation services in their working environment. We found that qualitative data from actors' field notes tended to confirm the quantitative ratings.

Our previous research showed that information from the patient coupled with visual and linguistic cues affected advisors' perceptions of the chances of quitting and hence the likelihood of recruitment into the service.[15 16] Previous studies also confirm that patient characteristics such as age, ethnicity and mental status may form barriers to engaging service users into smoking cessation interventions.[15 37] This evidence coupled with feedback from our pharmacy staff focus group-influenced recruitment of simulated clients from diverse backgrounds presenting a range of different scenarios. Beliefs and attitudes underlying prejudgements of treatment success were addressed in the STOP training.[16 24] This second pilot study suggests that STOP trained pharmacy staff engaged with all the actors regardless of age and ethnicity and the scenario presented.

The results of this pilot study have informed the STOP logic model in readiness for the STOP trial (figure 1). For example, site initiation visits and training sessions have been separated because these have a unique purpose and need to take place at different times and in different settings. The outputs in the logic model have been revised to reflect the elements of the intervention which are focused on staff behaviour, particularly initial client engagement, more clearly.

### Weaknesses

Pharmacy staff who chose to attend training may have had different baseline characteristics when compared with those who did not attend. We did not assess engagement skills before the STOP training intervention to gauge the base level of engagement with clients. Nevertheless, our findings are consistent with the suggestion that STOP training may improve pharmacy staff members' ability to engage clients in the NHS SSS. In the main trial, we plan to use the same method of fidelity assessment outlined here. However, we will include control pharmacies where staff have not been offered STOP training which will allow us to compare the performance of pharmacy staff who received STOP training and who did not to quantify the benefit arising from the intervention.

There was inconsistent reporting of SSS materials seen by simulated clients, particularly in rates of display for posters and the STOP study badge. There were instances where in the same pharmacy, some simulated clients reported seeing a poster or leaflets while others did not. This may reflect human error, in that there could have been posters up that some simulated clients simply did not see because they were distracted by the busy pharmacy environment. Another explanation could be that as

the simulated clients visited at different times, the materials were displayed on some days but not on others.

## Strengths and weaknesses in the context of other studies

Previous studies using simulated patients to assess community pharmacy staff performance have focused on the provision of over the counter medication,[29 32 33] whereas our research examines the expanded role of pharmacy staff as agents for health behaviour change. However, we used similar methods: covert visits, blinding to time and number of visits to minimise detection.[29 32] A key strength of our approach was the use of simulated clients with lived experience of smoking or smoking-related health conditions such as asthma. Moreover, feedback from staff in our study indicated no detection of our simulated visits while detections were reported in other studies.[29 33]

In this study, we assessed the initial interaction between pharmacy worker and smoker which usually takes place over the counter in community pharmacies. However, other researchers have audio-recorded consultations with stop smoking advisors allowing detailed examination of their interactions with clients, which take place in a dedicated/private consulting room.[8] Due to ethical considerations related to obtaining patient consent, it was difficult to audio-record naturalistic interactions with pharmacy users within the time limitations of this study. In the main trial, we aim to supplement data from simulated client visits with audiotaped follow-up consultations between stop smoking advisors and actual service users.

One study[30] with a similar aim and methodological approach to ours randomly assigned pharmacies to two different scenarios.[38] With our method, all pharmacies were assessed against all six scenarios thus examining a broader range of skills and allowing a more thorough evaluation of potential gaps in service delivery. In the main trial, we will use a balanced design where each pharmacy is exposed to each scenario which will allow a more rigorous comparison of the degree to which the intervention is implemented between different pharmacies.

## Implications for clinical practice and policy

The methods that we outline here could potentially be adapted to evaluate the effects of any training programme intended to modify the clinical practice of pharmacy staff. Given that government policy in the UK is to expand the range of clinical services provided in pharmacies,[39 40] methods to evaluate the effects of training may be useful in refining interventions and developing new training programmes.

## Unanswered questions and future research

The simulated clients noted adequate display of intervention materials in the pharmacies. However, certain materials were displayed more prominently than others. In the main trial, there is a need to evaluate carefully the use of intervention materials and to understand the reasons why certain materials are given more prominence and used more often than others. A better understanding of these factors may lead to the development of more effective intervention materials which are more likely to be available to potential users of the intervention.

**Author affiliations**
[1]Centre for Primary Care and Public Health, Barts and The London School of Medicine and Dentistry, London, UK
[2]Queen Mary University of London, London, UK
[3]Barts and The London School of Medicine and Dentistry, London, UK
[4]Centre for Primary Care and Public Health, Queen Mary University of London, London, UK
[5]Department of Medicine, California Northstate University, Elk Grove, California, USA
[6]Centre for Health Sciences, Barts and The London School of Medicine and Dentistry, London, UK

**Acknowledgements** The authors want to acknowledge Darush Attar who is a pharmacist, behaviour change specialist and national public health trainer specialised in smoking cessation, co-facilitated the STOP Intervention training with SJ and WYJ.

**Contributors** SJ: led development of the final version of the intervention and designed the fidelity assessment. WYJ and SJ: recruited sites, delivered the training and collected qualitative and quantitative data for this second pilot study. SJ, VM and TKY: analysed the data. LS: led development of the initial intervention for the first pilot study, and gave critical comments on the manuscript alongside WYJ, VM, RS, TKY, ST, CG and SE. SJ and RW: drafted the paper with input from all other authors. All authors have seen and approved the final manuscript.

**Funding** This is a second pilot study of the NIHR-funded STOP programme of which RW is the chief investigator and CG, SE and ST are co-investigators. NIHR Programme grant RP-PG-0609-10181.

**Competing interests** None declared.

**Patient consent for publication** Not required.

**Ethics approval** Ethical approval for the study was obtained from the Queen Mary Research Ethics Committee (Reference: QMREC1830a).

**Provenance and peer review** Not commissioned; externally peer reviewed.

**Data sharing statement** Repilot study data (with identifyinginformation removed) will be available following completion of the project on request from the study guarantor, Professor Robert Walton, r.walton@qmul.ac.uk.

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
