## [Reviewer comments · BMJ Open]

ARTICLE DETAILS

TITLE (PROVISIONAL)	Evaluating NHS stop smoking service engagement in community pharmacies using simulated smokers: fidelity assessment of a theory-based intervention
AUTHORS	Jumbe, Sandra; James, Wai Yee; Madurasinghe, Vichithranie; Steed, Liz; Sohanpal, Ratna; Yau, Tammy; Taylor, Stephanie; Eldridge, Sandra; Griffiths, Chris; Walton, Robert

VERSION 1 - REVIEW

REVIEWER	Tim Coleman University of Nottingham
REVIEW RETURNED	19-Oct-2018

GENERAL COMMENTS	This is a nice paper describing one of the steps involved in an important process - securing better engagement with SSS from smokers. It's very novel and provides a methodology which will interest other intervention developers and the same fidelity assessment methods are planned for a major NHS trial so I suspect this accompanying publication could become highly cited. The paper is well written and my comments are intended to help improve clarity so that the reader can enjoy it more. 1 Language used in 2nd line of abstract results in implies a significant difference but I can't see that a significance test was performed and largish SDs may mean that distribution of scores overlaps too much for this to be true. Does this line need re-wording? 2 Can the initial pilot be mentioned earlier in paper - I wondered what a repilot is? 3 The PPI section feels a little plonked in at the end of the methods section- can it be placed earlier? 4 Would it be possible for key numerical findings to be highlighted in the results section of the main manuscript? It's hard to get a feel for these by reading the text at the moment. 5 Can a few more details regarding derivation and content of the engagement checklist be provided in the text? I had to scroll to the end of the paper to get an impression of this instrument's contents. 6 Can you provide a little more detail about thematic analysis methods? Also providing a summary description of the number
---

	and nature of themes, prior to describing these, would help the reader gain an overview of study findings. 7 In the methods section it would be useful to give the reader an impression of the key phrases that were mentioned in the STOP training. The fact that actors noticed these being used a lot features prominently in the paper so it would be reassuring for the reader to know how distinctive these phrases are (e.g. they aren't phrases that one would expect to hear anyway).
--	--

REVIEWER	Hazel Gilbert UCL UK
REVIEW RETURNED	19-Oct-2018

GENERAL COMMENTS	Smoking cessation support in pharmacies is an underused opportunity to provide help to smokers. Pharmacists generally do not have sufficient training in the skills and knowledge needed to motivate and advise smokers to quit. This study represents an attempt to address this issue with a well thought out programme aiming to equip pharmacists and their staff with the skills needed to approach and discuss the behaviour with smokers. While the study has been adequately conducted, and is a good start to addressing the issue, the handling of the resulting data and the reporting of the results does not do it justice is insufficient. More and better analysis of the data is needed and the study has not been written and presented well. The Methods section is not well organised. The design, recruitment, sample size and procedure should be included. The study used simulated smokers-clients but this is, in various places referred to as actors and 'mystery shoppers'. The method is not explained anywhere. There is no information about the pharmacies (location, size, etc) and how they were selected. There is also some confusion about the number of pharmacies. Six pharmacies were recruited but five reported. It would be useful to have the Tables numbered in the order in which they appear in the report. Unnecessary details about training are included. Several changes to the study are mentioned but these were actually changes to the training times. These details given suggest that too few people attended the previous training, but no reference to this possibility is given. The details are not needed, it would be sufficient to say that 'changes were made to facilitate training'. There is a passing reference to the location of the pharmacies on page 4, but no other information is given. Again Figure 2 is referred to before Figure 1, referring to the stop logit model, but nowhere is this model discussed or explained. The actors – were they paid?
---

Presumably to meet the requirements of PPI, a focus group is mentioned, but there is no follow up of this. Do they mean that the questionnaire was informed by the focus group? Otherwise it does not seem relevant.

Results are confusing and meagre, and information is missing.

To claim that interaction with STOP trained staff were higher than non STOP trained staff figures and analysis are needed. Are the scores significantly higher. It looks so, but without more explanation of the analysis conducted it is not possible to make this claim.

How many times did the actors see the same person in the pharmacy?

Was there agreement between the actors for the same pharmacies?

Of the notes taken, this is not a thematic analysis. What themes were found and where are they documented. It would be better to make it a quantitative analysis, i.e. how many times things were mentioned by trained staff.

Discussion

The first sentence is an assumption, the analysis of the data is not sufficient to make this assumption.

There is not sufficient analysis of the data to draw the conclusion that 'STOP trained pharmacy staff engaged with all actors'.

The STOP logic model is mentioned again here but there is no explanation or discussion of this model. Again details about flipcharts and tar jars and desk calendars are details that do not tell the reader anything about the trial or the intervention.

Much of the discussion reads like a protocol for a study, leading me to wonder whether this was written as a study or a protocol for a new study.

Sometimes details are brought up in the discussion which are not mentioned previously e.g. actors with lived experience of smoking (only four of them were ex-smokers and this in fact could introduce bias).

Minor points

P3 line 17 the SSS offers other pharmacotherapies besides NRT.

Table 1: to what do the numbers of behaviour change techniques refer?

In my view this paper might, after major revision, be more suited to a specialist journal such as IJPP or one specialising in pilot studies. To organise the paper better the authors might look at a similar paper reporting an Australian study (referenced in the paper).

VERSION 1 – AUTHOR RESPONSE

Reviewer ID	Comment/Suggestion	Authors' responses
Reviewer #1		
	It's very novel and provides a methodology which will interest other intervention developers and the same fidelity assessment methods are planned for a major NHS trial so I suspect this accompanying publication could become highly cited. The paper is well written and my comments are intended to help improve clarity so that the reader can enjoy it more. 1 Language used in 2nd line of abstract results in implies a significant difference but I can't see that a significance test was performed and largish SDs may mean that distribution of scores overlaps too much for this to be true. Does this line need re-wording? 2 Can the initial pilot be mentioned earlier in paper - I wondered what a repilot is? 3 The PPI section feels a little plonked in at the end of the methods section- can it be placed earlier? 4 Would it be possible for key numerical findings to be highlighted in the results section of the main manuscript? It's hard to get a feel for these by reading the text at the moment. 5 Can a few more details regarding derivation and content of the engagement checklist be provided in the text? I had to scroll to the end of the paper to get an impression of this instrument's contents. 6 Can you provide a little more detail about thematic analysis methods? Also providing a summary description of the number and nature of themes, prior to describing these, would help the reader gain an overview of study findings. 7 In the methods section it would be useful to give the reader an impression of the key phrases that were mentioned in the STOP training. The fact that actors noticed these being used a lot features prominently in the paper so it would be reassuring for the reader to know how distinctive these phrases are (e.g. they aren't phrases that one would expect to hear anyway).	Thank you for your appreciation of the novelty of our work and your very helpful suggestions Comment acknowledged. Line has been reworded The initial pilot has now been moved to the background section outlining aims, objectives and findings. 'Repilot' refers to the second pilot. This has been revised throughout the paper PPI has now been placed earlier and integrated to reflect flow of process Acknowledged - more numerical clarity embedded in results Now embedded within Table 6, and detailed in the results sections. This has been revised following second reviewer's suggestion. Acknowledged and detailed in methods as suggested
Reviewer #2		
	Smoking cessation support in pharmacies is an underused opportunity to provide help to smokers. Pharmacists generally do not have sufficient training in the skills and knowledge needed to motivate and advise smokers to quit. This study represents an attempt to address this issue with a well thought out programme aiming to equip pharmacists and their staff with the skills	Thank you for your comments. We have addressed the concerns expressed below. Hopefully we have added clarity to the study context, as well as stylistic and methodological approaches utilised

needed to approach and discuss the behaviour with smokers. While the study has been adequately conducted, and is a good start to addressing the issue, the handling of the resulting data and the reporting of the results does not do it justice is insufficient. More and better analysis of the data is needed and the study has not been written and presented well.

The Methods section is not well organised. The design, recruitment, sample size and procedure should be included.

The study used simulated smokers-clients but this is, in various places referred to as actors and 'mystery shoppers'. The method is not explained anywhere.

There is no information about the pharmacies (location, size, etc) and how they were selected. There is also some confusion about the number of pharmacies. Six pharmacies were recruited but five reported.

It would be useful to have the Tables numbered in the order in which they appear in the report.

Unnecessary details about training are included. Several changes to the study are mentioned but these were actually changes to the training times. These details given suggest that too few people attended the previous training, but no reference to this possibility is given. The details are not needed, it would be sufficient to say that 'changes were made to facilitate training'.

There is a passing reference to the location of the pharmacies on page 4, but no other information is given.

Again Figure 2 is referred to before Figure 1, referring to the stop logit model, but nowhere is this model discussed or explained.

The actors – were they paid?

Presumably to meet the requirements of PPI, a focus group is mentioned, but there is no follow up of this. Do they mean that the questionnaire was informed by the focus group? Otherwise it does not seem relevant.

To claim that interaction with STOP trained staff were higher than non STOP trained staff figures and analysis are needed. Are the scores significantly higher. It looks so, but without more explanation of the analysis conducted it is not possible to make this claim.

How many times did the actors see the same person in the pharmacy?

Was there agreement between the actors for the same pharmacies?

Comment acknowledged. This section has been reordered extensively to aid clarity

This method is now explained under study design. The word 'actors' has been removed to avoid confusion

Details on pharmacies' recruitment process have been added

Apologies for the slight error. This has now been corrected

This section has been revised accordingly with separation between the initial pilot and this second pilot for clarity

Detail now added

This has been corrected

Yes they were – detail now added in manuscript

Details on how feedback from the PPI focus group guided the second pilot study have been clarified

The differences were significant but due to the small sample we are only presenting descriptive statistics

Each actor only visited each pharmacy on one occasion

Ratings for pharmacies were generally similar between actors (see Table 4 and 6) but no statistical analysis looking at agreement was done due to the small sample

Of the notes taken, this is not a thematic analysis. What themes were found and where are they documented. It would be better to make it a quantitative analysis, i.e. how many times things were mentioned by trained staff. Discussion The first sentence is an assumption, the analysis of the data is not sufficient to make this assumption. There is not sufficient analysis of the data to draw the conclusion that 'STOP trained pharmacy staff engaged with all actors'. The STOP logic model is mentioned again here but there is no explanation or discussion of this model. Again details about flipcharts and tar jars and desk calendars are details that do not tell the reader anything about the trial or the intervention. Much of the discussion reads like a protocol for a study, leading me to wonder whether this was written as a study or a protocol for a new study. Sometimes details are brought up in the discussion which are not mentioned previously e.g. actors with lived experience of smoking (only four of them were ex-smokers and this in fact could introduce bias). Minor points P3 line 17 the SSS offers other pharmacotherapies besides NRT. Table 1: to what do the numbers of behaviour change techniques refer?	Suggestion acknowledged and revised accordingly Comments acknowledged. Sentences in this paragraph have now been posed as suggestions This detail was added to reflect the changes made to the STOP Intervention materials from the 1st to 2nd pilot. Now removed for clarity Thank you for this comment. The discussion is structured in line with the journal's suggestion, highlighting study's strengths and limitations, but also importantly signposting to the main trial. Influence of smoking history on simulated client ratings is now detailed in the results. Sentence now revised Numbers refer to coding of behaviour change techniques from the Behaviour change taxonomy. This similar format was previously published in the initial pilot study
--	---

VERSION 2 – REVIEW

REVIEWER	Hazel Gilbert UCL London UK
REVIEW RETURNED	06-Jan-2019

GENERAL COMMENTS	The authors have extensively revised this paper and the result is a very readable clear description of the study.
---